# Cut Flower Characteristics and Growth Traits under Salt Stress in Lily Cultivars

**DOI:** 10.3390/plants10071435

**Published:** 2021-07-14

**Authors:** Yun-Im Kang, Youn Jung Choi, Young Ran Lee, Kyung Hye Seo, Jung-Nam Suh, Hye-Rim Lee

**Affiliations:** 1Floriculture Research Division, National Institute of Horticultural & Herbal Science, Rural Development Administration, Wanju 55365, Korea; lillium@korea.kr (Y.J.C.); leeyr628@korea.kr (Y.R.L.); seokh@korea.kr (K.H.S.); suhjn@korea.kr (J.-N.S.); 2Agricultural Bigdata Division, Rural Development Administration, Wanju 54875, Korea; leehr26@korea.kr

**Keywords:** lily, chlorophyll a fluorescence, canonical discriminant analysis, principal component analysis, salt tolerance

## Abstract

Salt stress is a major constraint of crop productivity because it reduces yield and limits the expansion of agriculture. This study investigated salt tolerance in 26 cultivars of cut lilies (*Lilium* hybrids) by examining the effect of salt stress on the growth and morphological characteristics of flowers and leaves and their physiological properties (chlorophyll a fluorescence). Salt stress significantly affected the growth and development of cut lilies. Canonical discriminant analysis indicates that the middle leaf width, number of flowers, first flower diameter, petal width, and chlorophyll a fluorescence were correlated with salt stress, whereas plant height, the middle leaf length, days to flowering, and sepal width were less affected by the stress. The cultivars examined were divided into three groups: Group 1 included the salt-sensitive cultivars, which failed to develop normal flowers; Group 2 included cultivars sensitive to salt stress but tolerant to osmotic stress; and Group 3 was the salt-tolerant group, which developed commercially valuable flowers. In conclusion, the cultivars contained a variable range of cut flower characteristics and growth traits that can be employed for lily breeding programs and as material for molecular mechanisms and signaling networks under salt stress.

## 1. Introduction

Cut lilies (*Lilium* hybrids) are one of the most important ornamental plants worldwide that are used as cut flowers or pot plants and in gardening because of their excellent horticultural characteristics [1,2,3,4]. Commercial cultivation of cut lilies began in the USA, the Netherlands, and France by developing interspecific hybrids from crossable wild lilies [4]. The Netherlands, which dominates the total world lily bulb trade, had a production area of 5280 ha in 2020 [4,5]. The Netherlands produces more than 600 cultivars of lily bulbs a year and exports them to other countries [6]. In South Korea, cut lilies are ranked third in ornamental production, following cut roses (*Rosa* × *hybrida*) and cut chrysanthemums (*Chrysanthemum* sp.), with a total cultivation area of 91 ha and an annual value of USD 11 million in 2019 [6]. Most lily bulbs are imported from the Netherlands and other countries, generating a total cost of about USD 3 million in South Korea [7].

Lily cultivation is accompanied with breeding of new lily varieties. There are Asiatic hybrid lilies, Oriental hybrid lilies, and interspecific hybrid lilies from Oriental–Trumpet interspecific hybridization and *L. longiflorum*–Asiatic interspecific hybridization in mainly cultivated cultivar assortments. The information about various cut lily bulbs can help in management and cultivar selection for cultivation. In addition, development of new lily cultivars suitable for domestic environmental conditions, especially salinity, is required [8,9].

Salinity is the main environmental factor accountable for crop productivity in many areas of the world, especially in arid and semiarid regions [10,11,12]. About 30% of the world irrigated lands that produce approximately one-third of the crops are affected by salinity [13,14]. In Korea, lily cultivation is carried out by protected cultivation [8], which is exposed to excessive input of fertilizer [15]. The contribution of a large amount of chemical fertilizer and compost increases salt accumulation in the rooting zone, and decreases the soil’s chemical and physical properties [16,17]. In addition, high salinity water frequently increases the accumulation of salts in soil [18]. These conditions may increase the negative effects of salinity [14,19,20].

Salt stress is a major abiotic stress contributing to the loss of plant yields and productivity [13,21,22]. Salt stress, mainly due to accumulation of toxic Na^+^ and Cl^–^ ions in plant tissues, causes osmotic and ionic stress in plants [22,23]. The osmotic effect of salinity stress contribute to reduced growth rate and alters developmental characteristics, such as root/shoot ratio and maturity rate, and ionic stress reduces leaf growth [12,22]. The ionic effects of salinity stress lead to an energy problem in plants due to a reduction in photosynthetic capacity and different metabolic functions [13,24].

Several strategies have been used to increase production in salt-affected areas, such as leaching, drainage [23], the application of inorganic or organic chemicals on plants, and selection and breeding of salt-tolerant lines or crop cultivars [25]. Many researchers have reported that the cultivation of tolerant genotypes and development of salt-tolerant cultivars is the most effective way to overcome stress [23,24,25,26,27,28]. Fortunately, diversity for salt tolerance has been found in many species [27]. Recent development of high-throughput omics analysis techniques helped to elucidate salt stress mechanisms and enhance salt stress adaptation or tolerance in plants by transgenic biotechnological methods [26,28,29].

In ornamental plants, salt stress reduces the yield and impairs the quality of cut flowers [30]. Some researchers have reported decreased stem height and leaf senescence in some cut lily cultivars and genotypes under salt stress [31], and salt stress was found to affect physiological activity; photosynthetic rate and antioxidant enzyme activity [32]. The present study was conducted to evaluate the effect of salt stress, such as that caused by NaCl, on the flower and growth traits of cut lilies and classify the variation in lily cultivars. Our main objective was to identify the cultivars for breeding programs and genetic studies, and offer information on stress-tolerant cultivars for growing cut lilies.

## 2. Results

### 2.1. Summary of Cut Lily Growth Characteristics under Salt Stress

Effects on plant height, length and width of the middle leaf, days to flowering, number of flowers, first flower diameter, and petal and sepal width were measured between NaCl treatment and cultivars (Table 1). The cut flower traits among the tested cultivars were significant (*p* ≤ 0.001). Salt treatment had a significant effect on plant height (*p* ≤ 0.001), days to flowering (*p* ≤ 0.001), the number of flowers (*p* ≤ 0.001), chlorophyll a fluorescence (*p* ≤ 0.001), middle leaf length (*p* ≤ 0.01), and diameter of the first flower (*p* ≤ 0.01), petal width (*p* ≤ 0.01), but not significant on width of the middle leaf and sepal width. The traits of days to flowering (*p* ≤ 0.001), the number of flowers (*p* ≤ 0.001), chlorophyll a fluorescence (*p* ≤ 0.001), and plant height (*p* ≤ 0.05) were affected significantly when considering the joint effect of NaCl treatment and genotype.

The plant height, length and width of the middle leaf, days to flowering, number of flowers, first flower diameter, petal and sepal width, and chlorophyll a fluorescence were analyzed using canonical discriminant analysis to discriminate differences between NaCl treatment and control (Table 2). Canonical discriminant functions are the canonical weights of the variables and provide information about each variable. Absolute values and signs of standardized canonical coefficients were used to rank variables. The middle leaf width (0.46), number of flowers (0.58), first flower diameter (0.44), petal width (0.31), and chlorophyll a fluorescence (1.01) were high values signifying salt stress, whereas plant height (0.24), the middle leaf length (0.16), days to flowering (−0.13), and sepal width (−0.01) were less affected.

### 2.2. Characterestics of Cut Lily Cultivars

The commercial groups of *Lilium* are Asiatic hybrids (A), LA interspecific hybrids (LA) from interspecific crosses between *L. longiflorum* and Asiatic hybrids, Oriental hybrids (O), and OT interspecific hybrids (OT) from interspecific crosses between Oriental hybrids and Trumpet hybrids. Significant differences were found in the cut flower traits of *Lilium* cultivars and groups between the salt treatment and control plants (Table 3). In Asiatic hybrids and LA interspecific hybrids, plant height generally decreased. There was no significant reduction in days to flowering, petal width, and sepal width. In addition, chlorophyll a fluorescence was not significantly altered by salt stress in Asiatic lily cultivars and LA hybrids. The flower number of Albufeira (A) (*p* ≤ 0.05) and flower diameter of Navona (A) (*p* ≤ 0.01) decreased significantly. In Eyeliner (LA), plant height (*p* ≤ 0.01) and length and width of the middle leaf (*p* ≤ 0.01) decreased under salt stress, but the number and diameter of flowers were similar to those in control plants. In Brunello (A) and Merluza (A), there was no difference in flower characteristics between plants subjected to salt treatments and the control.

Salt stress changed cut lily characteristics in Oriental hybrids and OT interspecific hybrids compared to Asiatic hybrids and LA interspecific hybrids. The flower numbers of Oriental hybrid lilies, Body Guard (O) (*p* ≤ 0.01), Monteneu (O) (*p* ≤ 0.01), and Sein (O) (*p* ≤ 0.001) were all eventually aborted, and commercial cultivation was not possible under salt stress. Most Oriental hybrids and OT interspecific hybrids showed an increment in days to flowering under salt stress, while declines were seen in Pink Mist (OT) (*p* ≤ 0.001), Sensi (OT) (*p* ≤ 0.001), and Cadenza (O) (*p* ≤ 0.001). Plant height of Oriental hybrids and OT interspecific hybrids generally deceased under salt stress, significantly in Siberia (O) (*p* ≤ 0.05), Pink Mist (OT) (*p* ≤ 0.05), and Zambesi (OT) (*p* ≤ 0.01). Compared to the control, salt treatment resulted in lower chlorophyll a fluorescence in most Oriental hybrids and OT interspecific hybrids.

### 2.3. Analysis of Principal Components in Cut Lily Cultivars

Principal component (PC) analysis was performed for each of the cut flower traits and coordinate of the cultivars with correlation matrix; the results are presented in Table 4 and Figure 1. To measure sampling adequacy and the possibility of the correlation matrix for PC analysis, Kaiser–Meyer–Olkin (KMO) and Bartlett’s test of sphericity were performed. The current study achieved a KMO value of 0.820, which indicated the sampling was adequate. Bartlett’s test scores less than 0.050 are favorable and suggest that significant relationships exist among variables. Bartlett’s significance level was 0.0001 in this experiment, thus confirming the appropriateness to perform principal component analysis. The main PCs were determined; the first and second PC had eigenvalues of 5.48 and 1.39. The first two components explained 76.4% of the average differences between salt stress and the control. For PC1 (60.9%), the traits of day to flowering, number of flowers, first flower diameter, and width of petal and sepal had high values. For PC2 (15.5%), the values for the traits, plant height, length and width of the middle leaf, and chlorophyll a fluorescence, were high. The cultivars were separated into three groups. Group 1 had a high value of PC1 and a low value of PC2 (Sein, Body Guard, and Monteneu). Group 2 had a moderate value of PC1 and a high value of PC2 (Zambesi, Siberia, Palazzo, Eyeliner, and X Factor). The last group, Group 3, contained the remaining cultivars with a moderate value of PC1 and a low value of PC2 (Brunello, Navona, Albufeira, Merluza, All Choice, Cadenza, Clear Water, Dynamite, Fenice, Gerona, Glendale, Kayenta, Lingerie, Patagonia, Universe, Pink Mist, Sensi, and Stentor). In the cultivars of Group 1, the growth of the cut lilies was reduced and all flower bud abortions were observed under salt stress. The cultivars of Group 2 showed decreased plant height, leaf length and width, and chlorophyll a fluorescence, but flowers in Group 2 developed normally compared to Group 1. In Group 3, there were few significant changes in the growth and flower traits.

## 3. Discussion

Cut lilies are generally produced in greenhouses using irrigation systems to guarantee optimal conditions [33,34]. Irrigation systems are particularly prone to salinization; irrigation with saline water and evaporation from the medium surface lead to the accumulation of salts [33,34,35,36,37]. Salinity affects many morphological, physiological, and biochemical processes, including seed germination, plant growth, and water and nutrient uptake [38,39,40]. This problem significantly reduces the value and yield of the crop. Strategies to avoid this problem are flesh water input [13]; the application of soil amendment such as biochar, organic matter, and mineral matter [41,42]; and admixture of the soil [20]; these are still prohibitively costly and require much effort on a large scale. Therefore, one of the most effective and feasible ways to sustain crop production under salt is to develop new cultivars and genotypes by enhancing their salt tolerance [13,43]. In the present study, we compared 26 cultivars belonging to four groups: Asiatic hybrids, LA interspecific hybrids, Oriental Hybrids, and OT interspecific hybrids, under 8 dSm^−1^ NaCl salinity conditions and control.

The results showed a significant difference among cultivars and salt stress in the characteristics and growth traits investigated (Table 1 and Table 2). Saline conditions decrease height, shoot length, leaf area, fresh weight, and dry weight in many plants. Under salt stress, water deficit and wilting occur because of rapid change in the osmotic potential difference between the plant and exterior environment [36,40,44,45]. Accompanying this water deficit stress are ABA biosynthesis and transportation throughout the plant initiating stomatal closure, among many other responses, and a decrease in photosynthetic pigments and photosynthetic capacity [46]. In the present study, the height of most cultivars decreased significantly, and leaf length was prone to decrease. A reduction in the salt-induced growth rate and a related decline in leaf area and plant height occurred [46]. Additionally, salt stress resulted in a decrease in the development rate of leaf area expansion due to the key role of water during leaf photosynthesis. Specifically, salt damage can shorten the elongation zone and/or period of the leaves, reducing the growth rate of leaves and the rate of local blade expansion [47,48].

A common plant response to salt stress is growth retardation and delayed flowering [49]. In cut lilies, salt stress changes days required to flowering, but this behavior is not exhibited by all cultivars. All Choice, Clear Water, Kayenta, Siberia, Universe, X Factor, Palazzo, and Zambesi increased days to flowering, but Cadenza, Sensi, and Pink Mist flowered earlier compared to control (Table 3). Several metabolites/metabolic pathways that contribute to stress acclimation also play a role in their development [50]. The susceptibility of plants to salinity stress is also because of changes in molecular programs that affect development. A number of genes (*DELLA*, *BFT*, *NTL8*, *TFL*) have been identified as being responsible for salinity-induced delays to flowering [51]. However, some plants, though flowering faster, complete their life cycle in the presence of salt stress [51].

The level of suppression of plant growth under salt stress differs in different plant organs [12]. The decline in the number of grains and flower diameter in Basmati rice [27] and the number of fruits in tomato (*Solanum lycopersicum*) [12] has been related to salt-stress susceptibility at the reproductive stage. Zeng et al. (2005) [52] reported that Na^+^ was the most significant parameter affecting growth performance, and shoot dry weight and number of tillers were more sensitive, in contrast to leaf area and plant height, in rice (*Oryza sativa* L.). Similar results were observed in the present experiment, where the number of flowers in all lily cultivars was more strongly correlated with salt stress than plant height or middle leaf length among growth characteristics.

Photosynthesis is one of the primary processes affected by salt stress [53]. Salt stress affects photosynthetic electron transport and inhibits the photosystem II (PSII) activity due to the accumulation of salts in chloroplasts [23]. Many researchers suggested the use of chlorophyll a fluorescence induction parameters to detect metabolic perturbations by abiotic stresses. Canonical discriminant analysis is a multivariate statistical technique that can identify differences among individuals and improve our understanding of the relationships among the variable measured [54]. Signs of standardized canonical coefficients are used to rank variables in order of their contribution. Standardized canonical coefficients larger than 0.3 divided by the squared root of the eigenvalue of canonical function I are considered large enough to contribute significantly to the classification [55]. In the present study, the quantum yield of PSII (*Fv*/*Fm*) had the highest canonical correlation coefficient (1.01). Then, standardized canonical coefficients of the canonical discriminant functions were weighted toward the number of flowers.

Using the concentration equivalent of electrical conductivity of the soil saturation extract, most plants suffer salt injury at 4 dS m^−1^ or higher [56]. In the present experiment, cut lily cultivars were exposed to 8 dS m^−1^ NaCl to induce high salt stress and changes in flower and growth traits of cut lily cultivars (Table 3). Increasing the nutrient solution salinity to 2.5 dS m^−1^ and 2.8 dS m^−1^ did not reduce the yield or cause damage in lilies [30,57]. A sequential increase in salinity to 4.5 dS m^−1^ improved the quality of cut lilies [30] but decreased the stem height, raceme length, and flower diameter [57]. Asiatic hybrids with *L. lancifolium* demonstrated greater resistance to stress [58]. Brunello has been used as a breeding material [59,60] for development of cultivation technology [61] and for the identification of physiological characteristics [62]. Merluza was adapted to environmental conditions in Beijing [9]. The Asiatic hybrid Brunello and *L. longiflorum* Asiatic hybrid Merluza are assumed to be tolerant to salt stress. Siberia, an Oriental hybrid, has high light energy conversion efficiency, as estimated from chlorophyll a fluorescence [63], and moderate resistance to chill stress [64]. With salt stress not affecting flower characteristics, number of flowers per plant, or flower diameter, Siberia is considered to be moderately tolerant. The Oriental hybrids All Choice, Dynamite, and Universe are considered tolerant to salt stress. The flowers of the Oriental Trumpet hybrid lilies Palazzo, Pink Mist, and Sensi preserved their commercial value under salt stress, although the levels of chlorophyll a fluorescence were significantly lower compared with those in the control. A similar study on Citrus reported there were different responses to cope with salinity: (1) maintenance of active photosynthetic system, and (2) rapid reductions of net photosynthetic rate [65].

Many researchers have defined the effects of salinity on plant growth as the result of water deficit due to low water potential, stress caused by toxic ions, and nutritional imbalance triggered by reduced nutrient uptake and transport [12,23,26,38,66]. The most valuable agronomical traits to discriminate cultivars under salt stress might represent the efficient characteristics [23]. Plants possess anatomical and morphological mechanisms of adapting and avoiding salinity, including tissue tolerance to osmotic stress, ion homeostasis, and detoxification [28,67]. PCA was used to investigate overall variation in data by means of linear relationships among measured variables under salt stress [68]. As shown in the PC analysis results, the cultivars were separated into three groups (Figure 1, Table 4). Group 1, which included salt-sensitive cultivars, failed to develop normal flowers under salt stress, whereas the salt-tolerant Group 3 produced commercially valuable flowers. In Group 2, decreases in plant height and length, width of the middle leaf, and chlorophyll a fluorescence were higher, rather than changes of flower-specific characteristics, number of flowers, and days to flowering.

The cultivars and germplasms contained a variable range of cut flower traits, which can be used in lily breeding programs and as material for studying the molecular mechanisms and signaling networks under salt tolerance. Plant responses to salt stress were divided into water stress effects, which cause metabolic changes in reproductive development by osmotic effect, and salt-specific effects, which cause leaf damage by increasing ion toxicity [12,36]. Group 1, including Sein, Body Guard, and Monteneu, which had no normal flowers, can be salt sensitive and susceptible to osmotic stress. Group 2, including Siberia, Palazzo, X Factor, Eyeliner, and Zambesi in which salt stress affected leaf growth more than that of the flowers, are tolerant to osmotic stress but sensitive to salt stress. Group 3, including Brunello, Merluza, All Choice, Dynamite, and Universe, etc., are tolerant in the long term to flowering. In conclusion, salt stress significantly changes cut lily growth and flower development in a cultivar- and trait-specific pattern. This result offers a screening criterion for improving the salt tolerance of cut lily cultivars. In addition, the relation between characteristics and growth traits under salt stress were analyzed. This research contributes to the understanding of the complex salt-tolerance mechanisms and development of markers for the evaluation of salt stress effects and selection of salt stress-resistant lilies.

## 4. Materials and Methods

### 4.1. Plant Materials and Growth Conditions

The experiments were conducted in a greenhouse of the National Institute of Horticultural and Herbal Science (NIHHS) in Jeonju, Korea. A total of 26 cut lily cultivars (*Lilium* spp.): Asiatic hybrid lilies (Brunello and Navona), *L. longiflorum* Asiatic hybrid lilies (Albufeira, Eyeliner, and Merluza), Oriental hybrid lilies (All Choice, Body Guard, Cadenza, Clear Water, Dynamite, Fenice, Gerona, Glendale, Kayenta, Lingerie, Monteneu, Patagonia, Seine, Siberia, Universe, and X Factor), and Oriental Trumpet hybrid lilies (Palazzo, Pink Mist, Sensi, Stentor, and Zambesi) were used for this experiment. Bulbs for cut flower production were purchased from Woori Seed Co. (Woori Seed, Gwacheon, Korea) who imports from the Netherlands. Imported bulbs were stored at −1 °C until planting, and defrosting was performed at 8 °C for 2 weeks for this experiment. Eight bulbs of each cultivar were planted in a container (30 cm (W) × 60 cm (L) × 20 cm (H)) filled with a commercial germination medium (Hungnong, Pyeongtaek, Korea) on 17 March 2017. Two liters of lily growing solution (Coseal, Seoul, Korea) were irrigated every week until the end of the experiment. Two containers were prepared for NaCl treatment and control.

### 4.2. Salt Treatment and Plant Growth Measurement

Plants were treated either with the lily growing solution alone (control) or amended with 2 L NaCl solution added to the container every week. The salinity of the solution was adjusted to 8 dS m^−1^ using a conductivity meter (CyberScan 100, Eutech, Singapore). The same amount of water without added NaCl was used as control. Salt treatment was initiated at the stage of visible sprouts. Four plants per treatment were randomly chosen to quantify the cut flower traits. When each cultivar reached flowering stage, different cut flower characteristic traits, namely, plant height, length and width of the middle leaf, days to flowering, number of flowers, diameter of the first flower, and petal and sepal width were measured at the flowering stage. Plant height was measured from the ground to the inflorescence top, and days to flowering were calculated from planting day to florescence of the first flower.

### 4.3. Determination of Chlorophyll a Fluorescence

To examine the effects of salt stress on plant physiological parameters, chlorophyll a fluorescence was measured at the flowering stage. The third leaf from the inflorescence was used for measurements plants in control and salt treatment plants. Chlorophyll a fluorescence of the leaf samples was measured using PAM-100 (Waltz, Effeltrich, Germany). The maximum photochemical quantum yield (*Fv*/*Fm*), maximal variable fluorescence (*Fv*), and maximal fluorescence intensity (*Fm*) were calculated to analyze the effect of salt stress. The leaves of four randomly chosen plants were shaded for 1 h, and then the leaves were removed for analysis in the laboratory.

### 4.4. Statistical Analysis

Analysis of variance (ANOVA) was conducted to analyze the effect of salt stress on cut flower traits, and the differences between treatment plants and the control for each cultivar were compared using t-tests. Canonical discriminant analysis was used to determine the magnitude and direction of salt stress on cut flower traits in all cultivars. CDA determines how to separate or discriminate the treatment of individuals, given quantitative measurements of several variables [54]. This approach distinguishes several uncorrelated canonical discriminant functions (CDF) or canonical variables [54]. CDFs are the canonical weights of the original variables and provide information about the discriminatory power of each variable. Absolute values and signs of standardized canonical coefficients are used to rank variables in order of their contribution and to characterize the function [54]. Consequently, the cultivars were assorted into tolerant and sensitive genotypes based on average differences in cut flower characteristics between treatment and control plants using principal component analysis (PCA). PCA generated principal components are linear combinations of the growth and flower characteristic variables. Before analyzing PCA, Kaiser–Meyer–Olkin and Bartlett’s test were performed [69]. The PCs summarize the maximum possible variation projected onto two dimensions [68,70]. The combined approach adopted in this study, based on the use of PCA and CDA, was used to identify major variables between the external criteria [54,70,71,72]. The analyses were performed using SAS (version 9.2; SAS Institute, Cary, NC, USA).

## Figures and Tables

**Figure 1 plants-10-01435-f001:**
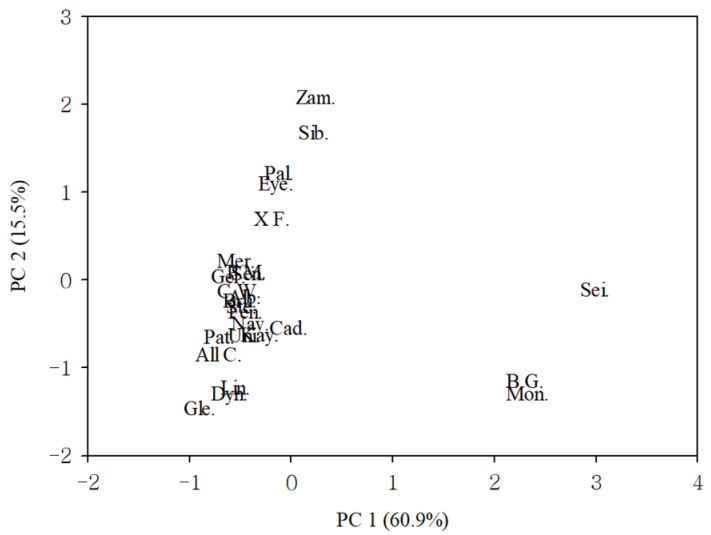
Distribution of lily cultivars according to multivariate factor analysis using the differences in the flower characteristic and the growth trait means, plant height, middle leaf length, middle leaf width, days to flowering, number of flowers per plant, first flower diameter, petal width, and sepal width in the control and plants subjected to NaCl treatment (8 dS m^−1^). Albufeira, Alb.; All Choice, All C.; Body Guard, B.G.; Brunello, Bru.; Clear Water, C.W.; Cadenza, Cad.; Dynamite, Dyn.; Eyeliner, Eye.; Fenice, Fen.; Gerona, Ger.; Glendale, Gle.; Kayenta, Kay.; Lingerie, Lin.; Merluza, Mer.; Monteneu, Mon.; Navona, Nav.; Pink Mist, P.M.; Palazzo, Pal.; Patagonia, Pat.; Seine, Sei.; Sensi, Sen.; Siberia, Sib.; Stentor, Ste.; Universe, Uni.; X Factor, X F.; Zambesi, Zam.

**Table 1 plants-10-01435-t001:** Summary of the correlation among the flower characteristic and the growth traits in cut lily cultivars under the control and NaCl (8 dS m^−1^) treatments based on ANOVA.

Variables	Plant Height	Middle Leaf Length	Middle Leaf Width	Days to Flowering	No. of Flowers Per Plant	First Flower Diameter	Petal Width	Sepal Width	*Fv*/*Fm*^1^
Cultivars (*n* = 26, C)	*** ^2^	***	***	***	***	***	***	***	***
NaCl treatment (*n* = 2; N)	***	**	ns	***	***	**	**	ns	***
C × N	**	ns	ns	***	***	ns	ns	ns	***

^1^ The ratio of maximal variable fluorescence (*Fv*) and maximal fluorescence intensity (*Fm*). ^2^, ns, *, **, and ***: not significant, and significant at *p* < 0.05, 0.01, and 0.001, respectively.

**Table 2 plants-10-01435-t002:** The flower characteristics and the growth trait variables associated with NaCl treatment (8 dS m^−1^) as determined by canonical discriminant analysis, standardized canonical coefficients, and pooled within-class correlations with the first canonical function.

Variables	Canonical Function I	
	Standardized Coefficient	Pooled Within-Class Correlations
Plant height	0.24	0.24
Middle leaf length	0.16	0.16
Middle leaf width	0.46	0.46
Days to flowering	−0.13	−0.13
Number of flowers per plant	0.58	0.58
First flower diameter	0.44	0.44
Petal width	0.31	0.31
Sepal width	−0.01	−0.01
*Fv*/*Fm* ^1^	1.01	0.94

^1^ The ratio of maximal variable fluorescence (*Fv*) and maximal fluorescence intensity (*Fm*).

**Table 3 plants-10-01435-t003:** The comparison of the flower characteristics and the growth trait means of cut lily cultivars exposed to NaCl treatment (8 dS m^−1^) and control conditions. Asiatic hybrids (A), LA interspecific hybrids from interspecific crosses between *L. longiflorum* and Asiatic hybrids (LA), Oriental hybrids (O), and OT interspecific hybrids from interspecific crosses between Oriental hybrids and Trumpet hybrids (OT).

Cultivars	Treatment	Plant Height (cm)	Middle Leaf Length (cm)	Middle Leaf Width (cm)	Days to Flowering (d)	Flowers Per Plant (no.)	First Flower Diameter (cm)	Petal Width (cm)	Sepal Width (cm)	*Fv*/*Fm*
Brunello(A)	Con.	74.7	11.6	1.3	53.0	4.3	14.1	3.5	2.3	0.608
	NaCl	65.2	11.6	1.3	53.0	3.8	14.2	3.0	2.2	0.552
Navona(A)	Con.	56.3	11.6	1.4	53.0	4.3	14.1	3.4	2.8	0.596
	NaCl	55.5	10.5	1.3	53.0	4.0	12.0 **^, 1^	3.2	2.6	0.589
Albufeira(LA)	Con.	66.9	8.8	1.6	58.0	4.3	10.7	3.5	3.0	0.624
	NaCl	54.9 *	8.7	1.5	58.0	3.0 *	10.9	3.4	2.9	0.622
Eyeliner(LA)	Con.	80.5	10.7	2.4	53.0	4.5	13.0	3.7	3.0	0.673
	NaCl	64.9 **	9.7 **	2.0 **	53.0	4.3	11.2	3.4	2.9	0.511
Merluza(LA)	Con.	85.5	13.2	1.9	53.0	3.5	13.1	4.4	3.4	0.649
	NaCl	73.7	13.0	1.8	53.0	3.5	13.7	4.3	3.3	0.584
All Choice(O)	Con.	121.9	17.6	4.0	71.0	6.0	11.9	3.6	2.1	0.554
	NaCl	128.7	18.7	3.9	75.0 ***	6.3	11.0	3.1	2.2	0.380 *
Body Guard(O)	Con.	92.0	17.2	4.3	75.0	5.5	13.9	3.5	3.0	0.305
	NaCl	78.2 *	15.9	4.0	-	0.0 **	-	-	-	0.237
Cadenza(O)	Con.	81.8	17.3	4.7	79.0	3.8	20.3	6.3	4.8	0.653
	NaCl	72.4	15.7	4.9	75.0 ***	3.0	17.6	5.1	4.1	0.598
Clear Water(O)	Con.	84.8	20.5	4.7	75.0	1.5	16.0	6.5	4.6	0.363
	NaCl	87.3	19.8	4.6	78.0 ***	1.5	15.6	6.4	4.6	0.132 ***
Dynamite(O)	Con.	61.5	12.2	4.0	75.0	5.3	12.7	3.5	2.3	0.625
	NaCl	60.0	12.4	4.2	75.0	4.0	11.6	3.6	2.3	0.584
Fenice(O)	Con.	68.0	16.0	4.2	75.0	3.0	15.5	5.1	3.7	0.522
	NaCl	66.1	15.7	3.8	75.0	2.7	15.5	4.9	3.0	0.580
Gerona(O)	Con.	70.0	13.8	4.3	75.0	2.8	14.6	4.9	3.4	0.543
	NaCl	70.1	13.7	4.0	75.0	3.3	15.0	4.8	3.4	0.420
Glendale(O)	Con.	71.2	13.1	3.6	75.0	3.0	16.7	3.9	3.1	0.443
	NaCl	71.4	14.5	3.5	75.0	3.0	14.6	3.9	3.2	0.548
Kayenta(O)	Con.	95.0	18.7	3.9	79.0	3.3	20.2	6.5	4.2	0.451
	NaCl	88.2	19.1	4.0	82.0 ***^, 1^	2.3	17.7	5.6	3.9	0.268 **
Lingerie(O)	Con.	74.8	14.2	5.4	75.0	2.0	16.4	4.9	3.2	0.556
	NaCl	65.7	15.8	5.6	75.0	1.0	15.1	4.7	3.0	0.421
Monteneu(O)	Con.	70.7	14.5	3.9	79.0	4.8	12.6	4.3	2.7	0.598
	NaCl	66.9	12.7	3.8	-	0.0 **	-	-	-	0.295 ***
Patagonia(O)	Con.	84.1	14.4	4.4	75.0	1.8	12.3	5.5	3.7	0.678
	NaCl	81.4	14.0	4.5	75.0	1.5	12.6	5.5	4.0	0.586 *
Seine(O)	Con.	65.7	14.9	4.0	82.0	5.5	14.7	4.4	3.5	0.496
	NaCl	50.1	11.5 **	3.6	-	0.0 ***	-	-	-	0.200 **
Siberia(O)	Con.	81.5	18.2	4.2	79.0	3.5	14.5	4.1	2.9	0.542
	NaCl	66.3 *	15.5	3.8	86.0 ***	2.7	12.8	3.6	2.3	0.287 **
Universe(O)	Con.	74.5	18.7	3.4	75.0	4.8	16.3	3.8	2.7	0.584
	NaCl	76.3	17.5	3.5	78.0 ***	3.3	16.7	3.8	2.8	0.467 **
X Factor(O)	Con.	77.3	15.7	3.3	75.0	3.0	16.7	4.2	3.0	0.591
	NaCl	71.8	14.1	2.9	78.0 ***	3.0	15.0	3.8	2.7	0.465 **
Palazzo(OT)	Con.	93.0	19.4	4.7	58.0	2.3	20.0	6.0	4.1	0.469
	NaCl	86.7	18.5	4.2	62.0 ***	2.3	18.4	5.4 **	4.1	0.151 ***
Pink Mist(OT)	Con.	91.8	19.3	3.7	69.0	4.7	14.7	4.5	3.6	0.640
	NaCl	82.2 *	18.9	4.0	65.0 ***	4.5	14.5	4.5	3.7	0.277 **
Sensi(OT)	Con.	110.6	19.7	3.2	69.0	3.5	17.7	3.9	3.2	0.563
	NaCl	97.1	19.7	3.2	65.0 ***	3.5	15.1	3.7	3.3	0.457 *
Stentor(OT)	Con.	90.2	17.7	3.6	69.0	2.5	15.2	5.4	4.1	0.535
	NaCl	75.4	17.4	3.7	69.0	1.5 *	13.2	5.9	4.1	0.533
Zambesi(OT)	Con.	91.6	17.5	4.5	65.0	1.5	17.3	5.3	4.5	0.572
	NaCl	65.9 **	16.1 *	4.0 *	69.0 ***	1.3	14.6	4.4 *	3.9	0.333 *

^1^, *, **, and ***: significant at *p* < 0.05, 0.01, and 0.001, respectively.

**Table 4 plants-10-01435-t004:** Principal component analysis for the five principal components of cut flower and growth traits from differences between salt treatment and control. PC1, PC2, PC3, PC4, and PC5 explained 60.9%, 15.5%, 8.8%, 7.7%, and 4.0% of the variation, respectively.

Trait	PC1	PC 2	PC 3	PC 4	PC 5
Plant height	0.16	0.47	0.49	0.69	0.17
Middle leaf length	0.29	0.41	−0.14	−0.04	−0.85
Middle leaf width	0.17	0.48	0.39	−0.70	0.26
Days to flowering	0.40	−0.24	−0.01	−0.01	0.09
No. of flowers per plant	0.39	−0.25	0.03	0.12	−0.05
No. of flowers per plant	0.41	−0.16	0.05	0.02	0.13
Petal width	0.41	−0.12	−0.08	−0.06	0.11
Sepal width	0.42	−0.10	0.08	−0.04	−0.01
*Fv*/*Fm*	0.17	0.47	−0.76	0.11	0.39

## Data Availability

Not applicable.

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
