# Peer review of "Cut Flower Characteristics and Growth Traits under Salt Stress in Lily Cultivars"

_plants, 2021, doi:10.3390/plants10071435_

Round 1

Reviewer 1 Report

In this manuscript, the authors described the effect of NaCl on phenotypic traits of 26 lily cultivars. Please find below my comments and suggestions.

  • In the “Results” section, interpretation of results and biological conclusions are missing, so they should be added.
  • The “Results” section should explain:
    • which cultivars belong to which group.
    • what characteristics define a specific group.
  • The introduction could include more information about the different cultivars, and explain why these 26 cultivars in particular have been chosen.
  • Line 212: “In cut lilies, salt stress changes days required to flowering, but this behavior is exhibited by all cultivars.” I think the results indicate that “this behavior is NOT exhibited by all cultivars”.
  • Figure 1: For a better legibility, overlapping names could be replaced by abbreviations, which could be explained in the figure legend.

Author Response

Thank you very much for your kind comments for this manuscript.
All of the points you mentioned have reflected. 

Sincerely

Reviewer 2 Report

This is an interesting and well done paper, for the most part, that will be of interest to those working to improve cut flower germplasm for salinity tolerance, and has some relevance beyond that specific research niche, ie. in finding a diversity of responses to salt, such as some flowering time lengthening and some shortening. Table 3 is of particular interest, showing how different varietals respond to imposed salinity in different ways and varying extents, and should serve as an effective starting point for a breeding program. 

There is one major issue that should be addressed in this manuscript before acceptance in this journal. The authors use CDA to group salinity-induced characteristics, and use PCA to group cultivars. The description of how they went about these analyses is insufficient in order for the reader to know what they did, why they selected those analytical methods, which specific settings within each analysis were chosen. When doing a PCA analysis, one can select principle components based on those that surpass an eigenvalue of 1, or based on a 'breakpoint' in a scree plot, and can be done with or without rotation (and rotation can be done with Varimax or several other means). Please include details of how the PCA analysis (and also the CDA) was preformed, and why the particular settings were selected. Also, the reader should know results of KMO and Bartlett's tests for sampling size. 

Please and read and cite these, or some other papers that do the same, descripting how CDA and PCA work within the context of hort sci:

Cruz-Castillo, J. G., Ganeshanandam, S., MacKay, B. R., Lawes, G. S., Lawoko, C. R. O., & Woolley, D. J. (1994). Applications of canonical discriminant analysis in horticultural research. HortScience29(10), 1115-1119.

Iezzoni, A. F., & Pritts, M. P. (1991). Applications of principal component analysis to horticultural research. HortScience26(4), 334-338.

Typically, CDA or PDA are performed in this type of work, not both. Please clarify why and how both are used here and serve different functions in the analysis.

Some smaller issues to consider:

Lines 49-50: This is unclear – are you implying that greenhouse-growing conditions are prone to salinity? (this is more clearly stated in the first line of your discussion)

Line 73: “, and salt stress affect physiological activity” explain

Methods:

Line 316-: 8dS/m (80mM) was used in the salinity stressed group. Why choose this level of salinity? (you get to this in the discussion, that it's higher than what's been tested before, but is it a relevant level for greenhouse managers?)

Salinity was imposed only after sprouting, meaning sprouts developed in non-saline conditions and then had to acclimate to the shift. How is this relevant to production? Wouldn't salinity either be present continuously, or ramp up steadily through the plants' life? Please discuss this.

Table 4: "No. flowers per plant" is duplicated

Author Response

(The authors gave the same response as above.)

Round 2

Reviewer 2 Report

These changes to the manuscript are excellent. I think it's now ready for publication in Plants.

Author Response

The authors are thankful to the reviewer for the insightful review, and for helping us improve our manuscript.

Sincerely.